# Conditional Density Estimation with Histogram Trees

**Lincen Yang**    **Matthijs van Leeuwen**
LIACS, Leiden University
Einsteinweg 55, 2333CC Leiden, The Netherlands
{l.yang, m.van.leeuwen}@liacs.leidenuniv.nl

## Abstract

Conditional density estimation (CDE) goes beyond regression by modeling the full conditional distribution, providing a richer understanding of the data than just the conditional mean in regression. This makes CDE particularly useful in critical application domains. However, interpretable CDE methods are understudied. Current methods typically employ kernel-based approaches, using kernel functions directly for kernel density estimation or as basis functions in linear models. In contrast, despite their conceptual simplicity and visualization suitability, tree-based methods—which are arguably more comprehensible—have been largely overlooked for CDE tasks. Thus, we propose the Conditional Density Tree (CDTree), a fully non-parametric model consisting of a decision tree in which each leaf is formed by a histogram model. Specifically, we formalize the problem of learning a CDTree using the minimum description length (MDL) principle, which eliminates the need for tuning the hyperparameter for regularization. Next, we propose an iterative algorithm that, although greedily, searches the optimal histogram for every possible node split. Our experiments demonstrate that, in comparison to existing interpretable CDE methods, CDTrees are both more accurate (as measured by the log-loss) and more robust against irrelevant features. Further, our approach leads to smaller tree sizes than existing tree-based models, which benefits interpretability.

## 1 Introduction

Conditional density estimation (CDE) is a crucial yet challenging task in modeling the associations between features and a *continuous* target variable, which has received a lot of research interest since the 1970s [44]. By modeling the full conditional distribution, CDE is useful when the datasets are multi-modal, heavily skewed, or heteroscedastic. As a result, it is widely applied in various fields, including genomics [8], astronomy [3], wind power forecasting [19], and computer networks [47].

CDE provides richer information for data understanding than regression, which only models the conditional mean. This makes CDE desirable for well-informed decision-making in critical areas, such as healthcare [32, 53, 14], which calls for interpretability.

Despite its importance, recent CDE research has predominantly focused on black-box models, such as neural networks [10, 55, 39, 46, 50] and tree ensembles [13]. In contrast, intrinsically interpretable models for CDE are understudied. Specifically, decision tree-based methods are largely neglected, with CADET [6] being the only existing method to the best of our knowledge. However, CADET's assumption of a Gaussian distribution for the target variable in each leaf node limits its ability to model complex conditional densities.

As a result, kernel-based models became the standard among 'shallow' models for CDE, including methods based on kernel density estimation (KDE) [44], and linear models with the basis functions chosen as Gaussian kernels [54, 11]. Nevertheless, kernel-based models are arguably less interpretable than decision trees: as conditions in decision trees are directly readable, they are comprehensible to

38th Conference on Neural Information Processing Systems (NeurIPS 2024).

humans without statistical expertise (e.g., individuals affected by data-driven decisions rather than professional data analysts).

To address these limitations, we propose the Conditional Density Tree (CDTree), a flexible tree-based CDE model in which each leaf node consists of a histogram model. For illustration, Figure 1 shows the CDTree learned from a dataset about personal medical costs with demographic features [24]. The figure visualizes the histogram models for the conditional densities on three selected leaf nodes, along with the unconditional density. The readable rules, extracted from the tree paths, and the histogram visualizations, make the results easily comprehensible to humans without statistical expertise. For instance, the difference in medical costs between smokers and non-smokers is evident when comparing the plots. The full results with histograms on all leaf nodes, as well as further descriptions of the dataset, are provided in Appendix E.

Learning a CDTree from data is a challenging task, as it requires simultaneously optimizing the decision tree structure and the number of bins for histograms on all leaf nodes. While often used for either task, cross-validation is too time-consuming if we need to search for the optimal number of bins for every possible node split. Thus, we formalize the learning problem using the minimum description length (MDL) principle [16, 41], which we briefly review in Section 4.1. Adopting MDL eliminates the need for tuning the hyperparameter, by cross-validation, for both regularizing the decision trees [5] and for choosing the number of bins for histograms.

Our main contributions are as follows. First, we introduce the CDTree for the CDE task and formalize the learning problem with the MDL principle. Second, we propose an iterative algorithm that can search the optimal histogram for all possible node splits. Third, we benchmark against a wide range of competitors and demonstrate that CDTree is highly competitive. Specifically, CDTree is more accurate than existing interpretable CDE methods (as measured by the log-loss). Meanwhile, CDTree has smaller tree sizes than other tree-based methods, which benefits interpretability. In addition, CDTree is extremely robust to irrelevant features, which is noteworthy as irrelevant features are known to harm the convergence rate for CDE [17]. Further, we argue that the (intrinsic) explanations of CDTree are trustworthy only if the CDTree is robust against irrelevant features.

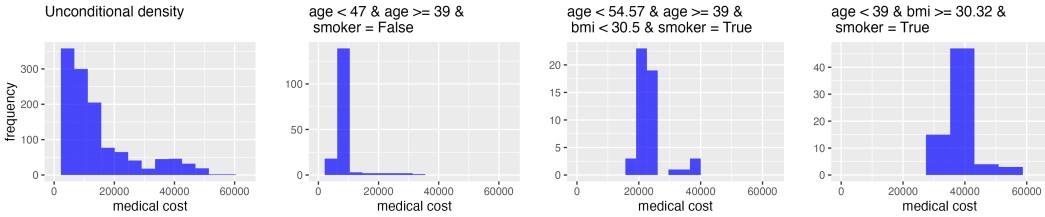

Figure 1: Three selected leaves from the CDTree modeling the conditional density of the medical costs given demographic features, together with the unconditional density for medical costs.

## 2 Related Work

**KDE-based CDE.** Several CDE methods have been developed based on kernel density estimation (KDE). The most straightforward approach, known as conditional-KDE (CKDE) [44], involves separately estimating the joint and marginal densities using unconditional KDE and then taking their ratio to obtain the conditional density. Another approach, $\epsilon$-neighborhood KDE (NKDE), estimates the conditional density by applying KDE to the subset of data points within the $\epsilon$-neighborhood of the target point, with the range of the neighborhood controlled by the parameter $\epsilon$.

However, KDE-based methods have several limitations. First, KDE is arguably less interpretable than tree-based models, as understanding KDE requires statistical knowledge, making it less accessible to domain experts and the general public. Additionally, while decision trees can naturally handle both discrete and continuous feature values, discrete features pose significant challenges for kernel-based methods. Specifically, CKDE requires estimating the joint density of the features and target variable, which necessitates the use of discrete kernels for discrete variables. These discrete kernels are often difficult to interpret [31]. As for NKDE, when both discrete and continuous feature values exist, the choice of the scale for the continuous variables unavoidably introduces a certain degree of arbitrariness in defining the distances used to characterize the neighborhood.

**Regression-based CDE.** Motivated by several issues of CKDE, including high variance as a plug-in estimator, the exponentially growing search space for bandwidth tuning, and the curse of dimensionality for estimating the joint density function, the method named least-squares CDE (LSCDE) [54] was proposed. LSCDE aims to directly estimate the ratio between the joint and marginal densities by assuming this ratio is a linear combination of several basis functions, chosen to be Gaussian kernels. Similarly, Fan et al. [11] proposed double-kernel local linear regression, which transforms the conditional density function into the conditional expectation of the (unconditional) density function of the target variable, leading to a least-squares approach as well. However, both methods face the challenge of bandwidth selection, as the value of the Gaussian kernel function must be calculated with the *entire* feature vector as input. This issue becomes particularly problematic as the search space for the bandwidth grows exponentially with the number of features.

**Tree-based CDE.** The only existing CDE method based on single trees that we are aware of is CADET [6], which improves CART regression trees [5] by designing a node-splitting heuristic specifically for CDE. However, CADET assumes a Gaussian distribution for the target variable on each leaf, which is far less flexible than our non-parametric histogram models.

**Black-box CDE.** Neural networks have been shown to perform well in a wide range of tasks. For CDE, the most well-known methods include NF [39] and MDN [4]. As for tree ensemble models, RFCDE [35] first fits a standard random forest, and then estimates the CDE by the weighted average of the unconditional density estimates obtained via KDE, with weights determined by the random forest. Furthermore, the recently proposed LinCDE [13] method learns a boosted tree model based on Lindsey's method [25]. While black-box models are highly accurate, we argue that interpretable CDE methods are valuable for applications in critical domains and for understanding the data.

## 3 Conditional Density Tree with Histograms

Tree-based models divide the feature space into disjoint (hyper-)boxes by recursively splitting on individual feature variables (for which we consider binary splits only). Consequently, the induced partition with $K$ subsets (leaves) can be represented as $M = \{S_k\}_{k\in[K]}$, where each leaf $S_k$ represents a subset of the feature space and $[K] := \{1, ..., K\}$. Further, each leaf is equipped with a single (unconditional) density estimator, denoted as $f_k(.)$. Thus, for a dataset $D = (x^n, y^n)$ with sample size $n$, the tree-based model $M$ first identifies the leaf to which each $(x, y) \in D$ belongs, and then estimates the conditional density $f(y|x)$ as $f_k(y)$ (assuming $x \in S_k$).

We choose histograms as our model class for each $f_k(.)$. Unlike previous parametric models [13, 6], which carry the risk of misspecification, histograms are non-parametric yet efficient. Additionally, in comparison to KDE, histograms do not require selecting a kernel or tuning the bandwidth.

We next describe our notations and formally present the histogram as a probabilistic model. Formally, a (fixed) histogram model partitions the domain of the target variable $Y$ into equal-width bins, and then approximates the density of $Y$ by piece-wise constants estimated from data. Thus, we can denote a histogram model as a tuple $H = (B_l, B_u, h)$, with $B_l$ and $B_u$ respectively representing the lower and upper boundary of the histogram, and $h$ the number of bins. A histogram model $H$ can parameterize a family of distributions by $\alpha = (\alpha_1, ..., \alpha_h)$, with the probability density function in the form of $f_H(Y = y) = \frac{\alpha_j}{(B_u - B_l)/h}$, $\forall y \in [B_l + (j-1)\frac{B_u - B_l}{h}, B_l + j\frac{B_u - B_l}{h})$, in which $(B_u - B_l)/h$ is the bin width and $[B_l + (j-1)\frac{B_u - B_l}{h}, B_l + j\frac{B_u - B_l}{h})$ denotes the interval for the $j$th bin. In practice, the histogram boundaries $B_l$ and $B_u$ are often set based on the dataset at hand, by prior knowledge, or by the range of the values plus/minus a small constant. Meanwhile, the parameters $\alpha$ can be estimated by the maximum likelihood estimator (i.e., by the empirical frequencies in each bin): $\hat{\alpha}_j = \frac{1}{n}\sum_{i=1}^{n} \mathbf{1}_{[B_l + (j-1)\frac{B_u - B_l}{h}, B_l + j\frac{B_u - B_l}{h})}(x_i)$, in which $\mathbf{1}(.)$ is the indicator function.

## 4 The MDL-optimal CDTree

We briefly review the minimum description length (MDL) principle, and then formalize the problem of learning a CDTree that consists of histograms as an MDL-based model selection problem.

## 4.1 Preliminary: the MDL principle for model selection

Rooted in information theory, the minimum description length (MDL) principle states that the optimal model is the one that compresses the data most [41, 15]. Precisely, given the dataset $D = (x^n, y^n)$, the MDL-optimal model is defined as

$$M^* = \arg \min_{M \in \mathcal{M}} -\log_2 P_M(y^n|x^n) + L(M), \tag{1}$$

in which $M$ is a CDTree and $\mathcal{M}$ the model class of all possible CDTrees for our learning task, meanwhile $L(M)$ is the code length (in bits) needed to transmit the model in a lossless manner. According to the Kraft's inequality [7], $L(M)$ can also be regarded as a prior probability distribution defined on the model class. Further, $P_M(.)$ is the so-called *universal distribution* [15]: as $M$ parameterizes a family of probability distributions, denoted as $P_{M,\theta}$, $P_M(.)$ can be regarded as a "representative" distribution such that the *maximum regret*, defined as $\max_{y^n}\{\max_\theta \log_2 P_{M,\theta}(y^n|x^n) - \log_2 P_M(y^n|x^n)\}$, can be bounded by $n\epsilon > 0, \forall \epsilon > 0$ as $n \to \infty$ [15], in which the maximum is defined over all possible values for $y^n$ in the domain of the target variable. Intuitively, the *regret* is the difference between the log-likelihood of $P_M(y^n|x^n)$ (which does not depend on $\theta$) and the maximum log-likelihood $\max_\theta \log_2 P_{M,\theta}(y^n|x^n)$. For our learning task, $\theta$ is the parameter vector that contains the histogram parameters (i.e., the $\alpha$'s) for all leaf nodes.

The MDL principle has been successfully applied to various data mining and machine learning tasks [12], and specifically to partition-based models, including histograms and classification trees/rules [37, 57, 36, 22]. The MDL framework provides a principled way of regularizing model complexity without any regularization hyperparameter to be tuned. Moreover, as the Bayesian marginal distribution is one specific type of the universal distributions, the MDL-based model selection can also be regarded as a generalization of Bayesian model selection [16].

## 4.2 Normalized maximum likelihood for CDTrees

The optimal universal distribution under the MDL framework is the so-called *normalized maximum likelihood* (NML) distribution, defined as [15, 49, 16]

$$P_M(y^n|x^n) = \frac{\max_\theta P_{M,\theta}(y^n|x^n)}{\int_{y^n} \max_\theta P_{M,\theta}(y^n|x^n)}, \tag{2}$$

under the condition that the denominator is finite. It can be shown that the NML distribution is the only distribution that leads to the minimax regret $\min_{P_M} \max_{y^n}\{\max_\theta \log_2 P_{M,\theta}(y^n|x^n) - \log_2 P_M(y^n|x^n)\}$, in which the denominator in Eq. 2 is exactly equal to the regret [15].

The denominator in Eq. 2 is in general prohibitively expensive to compute [15, 51, 43], with a few exceptions including the cases when the probabistic model represents categorical distributions [21], decision rules for classification [57], and one- and multi-dimensional histogram models [22, 28, 58]. We extend these previous results and prove that, for CDTrees with histogram models, the denominator (regret) is finite and is equal to the products of the regret terms of the NML distributions for one-dimensional histogram models, as shown in Proposition 1. This result is useful for efficiently calculating the denominator (regret) in Eq. 2 for CDTree models, as the regret terms for histogram models are known to be equal to those of categorical distributions [20, 21], for which an efficient algorithm exists with sub-linear time complexity [30].

**Proposition 1.** *Let $\theta = (\alpha^1, ..., \alpha^K)$ be the histogram parameters for histograms on all leaves, and let $\hat{\theta} = \arg\max_\theta P_{M,\theta}(y^n|x^n)$. Then $\int_{y^n} \max_\theta P_{M,\theta}(y^n|x^n) = \prod_{k \in [K]} \mathcal{R}(N_k, h_k)$, in which $\mathcal{R}(N_k, h_k)$ is the regret (denominator) of the NML distribution of the histogram model on the kth leaf node that contains $N_k$ data points and $h_k$ bins.*

We defer the proof to Appendix A due to space limitations.

## 4.3 Code length for the model

To encode a CDTree model in a lossless manner, we need to sequentially encode 1) the number of nodes in the decision tree, 2) the structure of the tree, 3) the splitting condition for each internal node in a predetermined order (e.g., depth first), 4) the number of bins for the histogram on each leaf

node, and 5) the boundaries $B_l$ and $B_u$ of the histograms. That is, let $L(.)$ denote the function that calculates code length in bits, the code length needed to encode a model $M$ can be decomposed to

$$L(M) = L(\text{\# leaves}) + L(\text{tree structure}) + \sum_{j \in [K-1]} L(\text{splitting condition of the } j\text{th } internal \text{ node})$$
$$+ \sum_{k \in [K]} L(\text{\# bins of histogram on the } k\text{th } leaf \text{ node}) + L(\{B_l, B_u\});$$

(3)

we next describe them in order.

**Encoding the tree size and structure.** As we consider full binary trees only, it is sufficient to encode the number of leaves $K$, which also determines the number of total nodes. As $K$ is a positive integer, we adopt the standard Rissanen's integer universal code [42], denoted as $L_\mathbf{N}(K)$, for which the code length is equal to $L_\mathbf{N}(K) = c + \log_2(K) + \log_2 \log_2(K) + ...$; the summation continues until a small enough precision is reached, and $c \approx 2.865$ is a constant.

Further, for full binary trees, the number of all possible tree structures for trees with $K$ leaves is equal to the Catalan number, denoted as $\mathcal{C}_{K-1}$ [52]. Hence, specifying one certain structure costs $\log_2 \mathcal{C}_{K-1}$ bits. Thus, $L_\mathbf{N}(K) + \log_2 \mathcal{C}_{K-1}$ bits are required for encoding the tree size and structure.

Note that while there exist multiple ways of encoding tree size and structure (e.g., one alternative is to leverage the joint probability of the tree size and the tree structure, which can be defined by treating the tree as a realization of a Galton-Watson process [33, 2]), our encoding scheme adheres to the principle of achieving *conditional minimax* [16] by explicitly putting a prior on the number of nodes.

**Encoding the splitting conditions.** An individual splitting condition on a single tree node consists of a variable name and a splitting value, which we encode sequentially.

To begin with, if the dataset contains $m$ feature variables, encoding the name of a certain variable $X$ costs $\log_2 m$ bits. Further, the code length needed for encoding the splitting value depends on the variable type. First, for a discrete variable $X$ with $|\mathcal{X}|$ unique values, specifying a single value cost $\log_2(|\mathcal{X}|)$ bits. Second, encoding the splitting value for a continuous variable can be achieved by sequentially encoding 1) the granularity level for the search space, denoted as the positive integer $d$, and 2) the exact value within the granularity level $d$. Specifically, with a fixed $d$, we consider as the search space the $C \cdot 2^{d-1}$ quantiles that can partition the values of $X$ into equal-frequency bins. Note that the values of $X$ are based on the subset of data points locally contained in this internal node.

Hence, encoding $d$ costs $L_\mathbf{N}(d)$ bits with Rissanen's code [42], and encoding one specific splitting value (quantile) costs $\log_2(C \cdot 2^{d-1}) = \log_2(C) + d - 1$ bits. That is, we treat the granularity level $d$ as a parameter to be optimized when learning a CDTree from the data, which avoids (arbitrarily) specifying the granularity in advance, a shortcoming in previous MDL-based methods (for other tasks instead of CDE) [22, 36, 58, 57, 28]. In contrast, the parameter $C$ can be used to express the prior belief about the hierarchical structure of the search space, as further discussed in Appendix C.3.

**Encoding the histograms.** One subtle choice we made is to set the boundaries for histograms on all leaves to be the same, and we set them as the global boundary of the target variable. This avoids unseen (test) data points falling outside the boundaries of the histograms on the leaf nodes. This is because while it may be common to assume that the boundaries for the histogram are known in *unconditional* density estimation, assuming the same for CDE is hardly realistic.

Therefore, the code length needed to encode the boundary $L(\{B_l, B_u\})$ in Eq. 3 is a constant that does not affect the model selection result. Consequently, it suffices to encode the number of bins for each histogram: for the histogram on the $k$th leaf with $h_k$ bins, it costs $L_\mathbf{N}(h_k)$ bits by Rissanen's integer code [42].

## 4.4 Model selection criterion

Combining Eq. 1, 2, and 3, together with the results from Proposition 1, we present the following MDL-score as our final model selection criterion:

$$M^* = \arg \min_{M \in \mathcal{M}} -\log_2\left(\max_\theta P_{M,\theta}(y^n | x^n)\right) + \sum_{k \in [K]} \log_2 \mathcal{R}(N_k, h_k) + L(M) \qquad (4)$$

which has the form of the regularized maximum likelihood, yet without the need to tune the regularization hyperparameter. Note that the exact form of $L(M)$ depends on the types of the feature variables; e.g., by assuming all variables are continuous, we obtain

$$L(M) = L_{\mathbf{N}}(K) + \log_2 \mathcal{C}_{K-1} + \sum_{j \in [K-1]} (\log_2(m) + L_{\mathbf{N}}(d_j) + \log_2(C) + d_j - 1) + \sum_{k \in [K]} L_{\mathbf{N}}(h_k),$$

where, as defined previously, $\mathcal{R}(.)$ denotes the regret term of each histogram, $L_{\mathbf{N}}(.)$ the Rissanen's integer universal code [42], $m$ the number of feature variables, $d_j$ the granularity level for the search space for the splitting values corresponding to the $j$th internal node, $C$ the constant that controls the hierarchical structure of the search space of the splitting values, and $h_k$ the number of bins for the histogram on the $k$th leaf node.

## 5   Algorithm

While finding the optimal tree-based model with the branch-and-bound approach is possible for regression [59] and classification [18], this does not apply to our task for two reasons. First, our MDL-based regularization term differs from traditional penalty terms based on tree size. Second, our task resembles optimizing *model trees* [27, 38] rather than classification and regression trees, as we aim to find the optimal histogram for each candidate split.

---

**Algorithm 1** Learn CDTree from data

---

**Input:** Training dataset $D$
**Output:** CDTree $M$

1 $M \leftarrow \{S_0\}$ ;                                              // One leaf node only
2 **while** *True* **do**
3    **for** $S \in M$ **do**
4       Search the condition that splits $S$ into two nodes and minimizes the MDL-score (Eq. 4). ;
      // Described in detail in Algorithm 2
5    **end**
6    **if** *Splitting any $S \in M$ cannot further decrease the MDL-score* **then**
7       **return** *CDTree $M$*
8    **else**
9       Among all $S \in M$, find the single $S^*$ to be split that minimizes the MDL-score.
10       Update $M$ by replacing $S^*$ with its two child nodes that minimize the MDL-score.
11    **end**
12 **end**

---

**A greedy approach for tree construction.** We thus take a heuristic approach to optimize our MDL-score in Eq. 4. As summarized in Algorithm 1, we start with a tree with one leaf node $M = \{S_0\}$; next, we iteratively update $M$ by replacing one of the leaf nodes with its 'best' two child nodes. Specifically, to achieve the lowest MDL-score at each iteration, we simultaneously search for 1) which node to split, 2) the splitting condition for that node, and 3) the optimal number of bins for the histograms. That is, we iterate over all leaf nodes in $M$; for each leaf node, we search for the splitting condition, along with the number of bins for the histograms on the (potential) child nodes, which as a whole minimizes the MDL-score. Notably, while an exhaustive search for the 'best' models on all potential child nodes is considered infeasible in traditional *model tree* methods [27, 38], we empirically demonstrate in Section 6.5 that our algorithm is comparable to KDE-based methods.

**Finding the child nodes.** We next elaborate on the search for the tree-splitting condition at each iteration, for which the pseudo-code is provided in Algorithm 2. For simplicity, we assume that all feature variables are either continuous or binary (i.e., categorical features are one-hot encoded in our implementation).

Further, we iterate over all columns of the feature matrix. For each column, we start with the granularity level $d = 1$ and generate candidate split points as the $C \cdot 2^{d-1}$ quantiles that lead to equal frequency binning of the values. Note that the quantiles are generated based on the subset of data points covered by the node to be split, rather than the entire dataset. We search for the best split point

at the fixed $d$ that yields the minimum MDL score among all generated candidates. We then proceed to the next granularity level $d + 1$ and repeat the process, which stops if no split point in the next granularity level results in a better (smaller) MDL score.

Last, we also need to search for the optimal number of bins for the histogram that leads to the minimum MDL score, which we defer to Appendix B.

---

**Algorithm 2** Find the best split for node $S$

---

**Input:** Training set $D_S = (x^{\{S\}}, y^{\{S\}})$ covered by node $S$, with $x^{\{S\}} = (\vec{x}_1, ..., \vec{x}_m)$ having $m$ columns, and constant $C$
**Output:** The splitting condition of $S$ that minimizes the MDL-score

13   **for** $j \in \{1, 2, ..., m\}$ **do**
14     $d \leftarrow 1$
15     **while** *True* **do**
16       $candidate\_splits \leftarrow$ The $C \cdot 2^{d-1}$ quantiles for equal-frequency binning for $\vec{x}_j$
17       **for** $s \in candidate\_splits$ **do**
18         $D_l \leftarrow \{(x, y) \in D_S | x_j \leq s\}$ ;       // Always $s = 1/2$ for binary features
19         $D_r \leftarrow \{(x, y) \in D_S | x_j > s\}$
20         Construct histograms for $D_l$ and $D_r$ that minimize the MDL-score conditioned on the fixed $j$ and $d$. ;       // Described in detail in Algorithm 3
21         Score $\leftarrow$ Calculate the MDL-score assuming $D_S$ is split into $D_l$ and $D_r$
22       **end**
23       **if** *The best Score is worse than that of the previous $d$* **then** **Break; else** $d \leftarrow d + 1$ ;
24     **end**
25     Record the best tuple $(d, s)$ for this column index $j$
26   **end**
27   **return** *The tuple $(j, d, s)$ that minimizes the MDL-score*

---

## 6 Experiments

We present our experiments to demonstrate the empirical performance of CDTree from various perspectives. Specifically, we aim to answer the following research questions: 1) Does CDTree provide more accurate conditional density estimation compared to existing interpretable (tree-based and kernel-based) methods? 2) As a proxy for interpretability, does CDTree have smaller tree sizes in comparison to other tree-based models? 3) Is CDTree robust against irrelevant "noisy" features? 4) Are the runtimes of our algorithm comparable to those of kernel-based methods?

### 6.1 Experiment setup

We use 14 datasets with numerical target variables from the UCI repository [1]. These datasets, summarized in Table 1, cover a wide range of sample sizes and dimensionalities. We benchmark our method against a variety of competitors, with all results obtained on the test sets using five-fold cross-validation. Our competitors include 1) NKDE and CKDE [44, 46], which are based on kernel density estimation, with bandwidth and $\epsilon$ (for NKDE only) tuned by cross-validation; 2) LSCDE [54], which directly estimates the ratio of joint and marginal density using linear models with Gaussian kernel basis functions; 3) tree-based method CADET [6], which fits a Gaussian distribution on each leaf node; 4) two more tree-based baselines introduced by us, CART-k and CART-h, which respectively fit a KDE model and a histogram after a CART regression tree [5] is learned from data.

Furthermore, we compare against three black-box models as "upper" baselines, including 1) neural network methods NF [39] and MDN [4], which apply both dropout and noise regularization [45], and 2) the recently proposed tree boosting method LinCDE [13]. Nonetheless, our goal with CDTree is *not* to be more accurate than black-box models, but to introduce a CDE method that is both interpretable and accurate.

For reproducibility, we provide further details about implementation and parameter choices in Appendix C. We made our source code public: `https://github.com/ylincen/CDTree`.

Table 1: Datasets from the UCI repository, with the numbers of rows and columns

| dataset | energy | synchronou | localizati | toxicity | concrete | slump | forestfire |
|---|---|---|---|---|---|---|---|
| # rows | 9568 | 557 | 107 | 908 | 1030 | 103 | 517 |
| # cols | 5 | 5 | 6 | 7 | 9 | 10 | 13 |
| dataset | navalpropo | skillcraft | sml2010 | thermograp | support2 | studentmat | supercondu |
| # rows | 11934 | 3338 | 2764 | 1003 | 9105 | 395 | 21263 |
| # cols | 17 | 19 | 21 | 26 | 27 | 44 | 82 |

Table 2: Negative log-likelihoods (smaller is better) on test sets. The best results among interpretable methods are shown in **bold**, and the best results among all interpretable and black-box models are marked by underlines. The datasets are ordered by their numbers of columns (ascending).

| | Interpretable models | | | | | | | Black-box models | | |
|---|---|---|---|---|---|---|---|---|---|---|
| Datasets | CADET | CART-h | CART-k | CKDE | LSCDE | NKDE | *Ours* | LinCDE | MDN | NF |
| energy | 3.55 | 3.09 | 3.06 | **2.47** | 3.38 | 3 | 2.93 | 2.93 | 2.78 | 2.86 |
| synchrono | -2.93 | -1.63 | -1.86 | **-3.59** | -1.25 | -1.57 | -2.11 | -1.85 | -2.94 | -2.64 |
| localizat | -0.23 | -0.55 | -0.01 | -0.26 | -0.61 | -0.28 | **-0.66** | -0.95 | -0.68 | -0.43 |
| toxicity | 1.8 | 1.5 | 1.38 | **1.32** | 1.34 | 1.55 | 1.53 | 1.29 | 1.24 | 1.23 |
| concrete | 4.17 | 3.75 | 3.93 | **3.32** | 3.66 | 3.91 | 3.72 | 3.47 | 2.97 | 3.18 |
| slump | 3.42 | 3.55 | 3.43 | **2.35** | 2.91 | 3.08 | 3.34 | 2.98 | 2.23 | 2.39 |
| forestfir | 134 | 3.96 | 4.39 | 4.85 | 4.68 | 5.55 | **3.43** | 4.35 | 3.26 | 3.23 |
| navalprop | -3.53 | -3.3 | **-3.66** | -2.8 | -2.88 | -3.19 | -3.6 | -3.36 | -4.12 | -3.75 |
| skillcraf | 94.4 | 0.46 | -0.42 | 1.54 | 1.61 | 1.56 | **-1.02** | 1.26 | 0.35 | 1.11 |
| sml2010 | 6.52 | 2.85 | 2.89 | **1.61** | 3.14 | 3.12 | 2.7 | 2.97 | 2.15 | 2.61 |
| thermogra | 2.21 | 0.66 | 0.72 | 0.66 | 0.94 | 0.94 | **0.64** | 0.59 | 0.56 | 0.52 |
| support2 | 97.3 | 0.51 | 0.32 | 2.09 | 2.46 | 2.13 | **0.29** | 1.48 | 1.53 | 1.24 |
| studentma | 3.83 | **2.65** | 2.66 | 2.89 | 4.19 | 3.11 | 2.66 | 2.59 | 3.85 | 3.54 |
| supercond | 9.6 | 3.84 | 4.36 | 4.55 | 4.17 | 4.19 | **3.48** | 3.87 | 3.33 | 3.5 |
| rank (all) | 8.79 | 5.68 | 6.04 | 5.11 | 7.46 | 7.68 | 4 | 4.46 | **2.57** | 3.21 |
| rank (intp.) | 6.07 | 3.43 | 3.86 | 3.14 | 4.57 | 4.86 | **2.07** | — | — | — |

## 6.2 Conditional density estimation accuracy

In Table 2, we report the average negative log-likelihoods (NLL) on the test sets, which can be regarded as an approximation to the expected log-loss $E(-\log(y|x))$, the standard loss function for measuring CDE accuracy.

**Comparison to interpretable models.** The NLL of the CDTree models are the best (lowest) in 6 out of 14 datasets among all interpretable models, which are shown in bold in the table. Further, among all interpretable models, our average rank is the best, as reported in the (last row) of Table 2. As the datasets are *increasingly* ranked based on their dimensionalities, we observe that CKDE—although also achieving the best NLL in 6 datasets—only performs well on datasets with low dimensions. Additionally, the other two kernel-based methods LSCDE and NKDE have in general worse accuracy.

Further, CDTree has better accuracy (lower NLL) than all tree-based competitors on almost all datasets (13 out of 14 datasets for CADET, and 12 out of 14 datasets for CART-h and CART-k). The superiority of CDTree highlights the advantages of 1) using non-parametric histogram models, and 2) conducting an exhaustive search for optimal histograms for all node splits when learning the tree structure, rather than fitting the model after the tree structure is fixed, as in CART-h and CART-k.

**Comparison to black-box models.** Neural network models generally exhibit better accuracy than interpretable models, as indicated by their average ranks. However, their non-transparency limits their applicability in critical areas. We argue that studying interpretable models and introducing CDTree paves the way for developing local surrogate models [40, 26], an essential approach for generating post-hoc explainability for black-box models, as no such method currently exists for CDE.

Further, the performance of CDTree is surprisingly on par with that of LinCDE. Since CDTree is based on a single tree while LinCDE is a tree ensemble model, we conjecture that the on-par performance is caused by the fact that CDTree adopts non-parametric histograms whereas LinCDE takes a parametric approach (with a much more flexible model class than Gaussian though).

We also report the standard deviations of NLL for all methods in Appendix D.1. The standard deviations for our method remain low, indicating stable performance across different datasets.

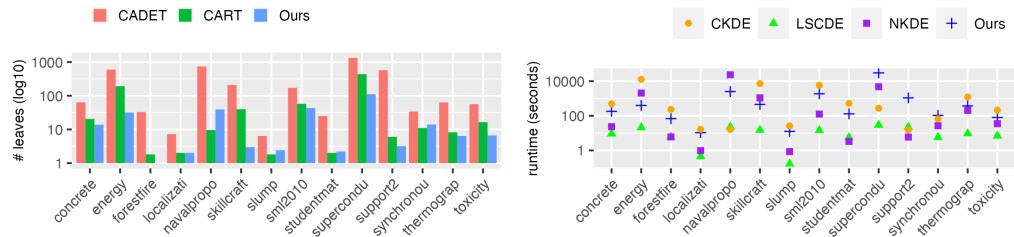

Figure 2: Left: the number of leaves for tree-based methods. Right: Runtimes of CDTree and kernel-based methods. Note that the y-axes are scaled by $\log_{10}(.)$

## 6.3 Complexity of trees

For tree-based models, the size of the trees is an important proxy for the degree of interpretability [29]. We hence compare the number of leaves of CDTree against the other tree-based models CADET and CART (note that CART-h and CART-k have the same tree structures). We demonstrate in Figure 2 (left) that CDTree has smaller tree sizes than the competitors on most datasets.

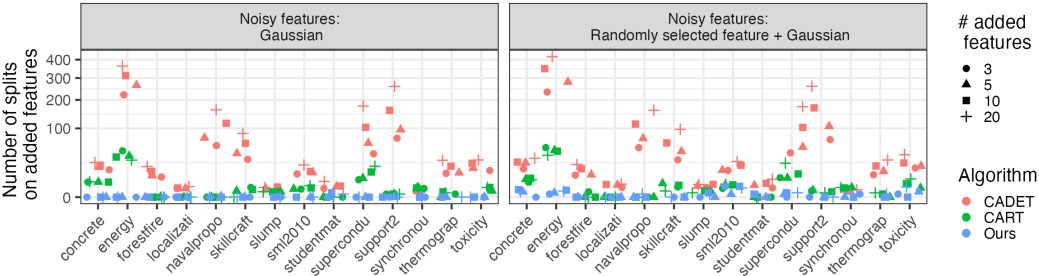

Figure 3: Number of internal nodes with split conditions that contain irrelevant features. The y-axis is scaled by the squared-root for better visualization.

## 6.4 Robustness to irrelevant features

We next investigate whether CDTree is robust against irrelevant features, which is particularly important for the interpretability of tree-based models, since the (intrinsic) explanation contained in the CDTree would not be trustworthy if such robustness did not hold.

Specifically, we generate 'noisy' features in two different ways. The first way is to add $w$ irrelevant features randomly drawn from the standard Gaussian distribution to each dataset. The second way is to first randomly select $w$ features from each dataset; then, for each selected feature $X_j$, we generate a 'noisy' feature by adding a Gaussian noise to it, i.e., $X_j' = X_j + N(0, s(X_j)/2)$, where $s(X_j)$ denotes the estimated standard deviation. We refer to the irrelevant features generated by the first (second) approach as independent (dependent) noisy features, where $w \in \{3, 5, 10, 20\}$ in both cases. Note that adding $X_j'$ to the dataset does not change the true conditional density as the target variable is conditionally independent of $X_j'$ given the original feature $X_j$.

Next, we train the tree-based models on expanded datasets that include irrelevant features. We count the number of nodes with splitting conditions that involve these added irrelevant features. As demonstrated in Figure 3, the number of splits on irrelevant features for our CDTree is almost always zero for both ways of generating noisy features. In contrast, the tree-based models learned by CART and CADET contain many more nodes with irrelevant features.

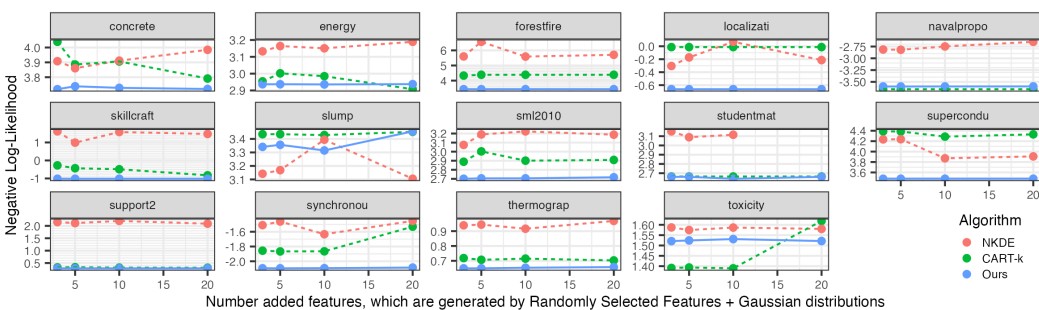

Figure 4: Negative log-likelihoods with different number of irrelevant 'dependent noisy' features. The results of CDTree (shown in blue lines) are stable on all datasets.

We also examine whether the negative log-likelihoods (NLL) remain stable when irrelevant features are added. For 'dependent noisy' features, the results are shown in Figure 4, where the NLL of CDTree remains nearly identical regardless of the number of added features—the only exception being the dataset "slump" (with only 103 rows). In contrast, the NLLs obtained by competitor methods are much less stable: the NLL of NKDE varies for most datasets, and CART-k shows visible changes in 6 out of 14 datasets. Similar results are observed when 'independent noisy' features are added, as shown in Appendix D.2.

## 6.5    Runtimes

The idea of fitting separate models on the leaves of a decision tree has existed for a long time [38]. Nevertheless, it is (still) often believed that fitting separate models for all possible node splits when growing a tree is infeasible. However, we show in Figure 2 (right) the runtime of CDTree and demonstrate that its runtimes are in general lower than the runtimes of CKDE (which requires intensive parameter tuning). While NKDE and LSCDE are in general faster than CDTree, the accuracy of their conditional density estimates are sub-optimal, as discussed previously. We exclude the comparison with other tree-based methods whose implementations are based on CART, as CART is highly optimized and known to be extremely fast. Further, as these tree-based methods either model the conditional means only (CART-h and CART-k) or assume a Gaussian model (CADET), they have far smaller search space, and hence are fast but less accurate.

## 7    Discussion

In this paper, we studied the interpretable conditional density estimation (CDE) models. Motivated by the fact that tree-based methods are arguably more interpretable than kernel-based methods yet have been largely disregarded for interpretable CDE, we introduced the Conditional Density Tree (CDTree). We formalized the learning problem under the MDL framework, proposed an iterative algorithm, and demonstrated its competitive empirical performance on a wide range of datasets.

**Limitations.** As histograms are used in the CDTree, we implicitly assume that the support of the target variable is bounded. Hence, the boundaries for the histograms need to be chosen in an ad-hoc way, possibly based on prior knowledge. Further, in practice, it may happen that the unseen data points fall outside the histogram boundary, for which the predicted (conditional) density will be 0, unless the full model is re-trained. We realize that this may cause some issues when using CDTree in practice, but meanwhile, we argue that 1) this is a limitation of the histogram model itself, 2) similar issues could happen in modelling quantiles by the empirical cumulative probability function, as in well-known methods like quantile regression trees and forest, and 3) all probability models have their own (implicit) assumptions on the probability tails, including those specifically designed for modeling the extreme value distributions [9, 23].

## Acknowledgement

We thank the anonymous reviewers who have provided constructive and valuable reviews. We also thank Prof.dr. Alexander Marx (TU Dortmund) for his feedback in developing the initial idea.

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

# A  Proof

**Proposition 1.** *Let* $\theta = (\alpha^1, ..., \alpha^K)$ *be the histogram parameters for histograms on all leaves, and let* $\hat{\theta} = \arg\max_\theta P_{M,\theta}(y^n|x^n)$. *Then* $\int_{y^n} \max_\theta P_{M,\theta}(y^n|x^n) = \prod_{k\in[K]} \mathcal{R}(N_k, h_k)$, *in which* $\mathcal{R}(N_k, h_k)$ *is the regret (denominator) of the NML distribution of the histogram model on the kth leaf node that contains* $N_k$ *data points and* $h_k$ *bins.*

*Proof.* Consider the dataset $D = \{x^n, y^n\}$ and a CDTree with $K$ leaf nodes $M = \{S_1, ..., S_K\}$, each leaf $S_k$ equipped with a histogram model $H_k = (B_l, B_u, h_k)$, in which $h_k$ denotes the number of bins. Denote the histogram density estimator associated with each $H_k$ as $f_{H_k}(.)$, and in short $f_k(.)$, which contains a family of distributions parameterized by $\alpha^k$. Further, denote the subset of data points that end up in the $k$th leaf as $D_k = (x^{\{k\}}, y^{\{k\}}) := \{(x, y) \in D | x \in S_k\}$. Thus,

$$
\begin{aligned}
\int_{y^n} \max_\theta P_{M,\theta}(y^n|x^n) &= \int_{y^n} \max_\theta \prod_{k\in[K]} f_k(y^{\{k\}}|x^{\{k\}}) \\
&= \int_{y^n} \prod_{k\in[K]} \max_{\alpha^k} f_k(y^{\{k\}}|x^{\{k\}}) \\
\text{(Fubini's Theorem)} &= \int_{y^{\{1\}}} \int_{y^{\{2\}}} ... \int_{y^{\{K\}}} \prod_{k\in[K]} \max_{\alpha^k} f_k(y^{\{k\}}|x^{\{k\}}) \\
\text{(re-arranging)} &= \prod_{k\in[K]} \int_{y^{\{k\}}} \max_{\alpha^k} f_k(y^{\{k\}}|x^{\{k\}}) \\
&= \prod_{k\in[K]} \mathcal{R}(N_k, h_k)
\end{aligned}
\tag{5}
$$

because the NML distribution for $D_k$ given the histogram $H_k$ is equal to [22]

$$
P_{H_k}(y^{\{k\}}|x^{\{k\}}) = \frac{\max_{\alpha^k} f_{\alpha^k}(y^{\{k\}}|x^{\{k\}})}{\int_{y^{\{k\}}} \max_{\alpha^k} f_{\alpha^k}(y^{\{k\}}|x^{\{k\}})}.
\tag{6}
$$

Further, it has been shown that the denominator (regret) of the NML distributions for histogram models is a function that only depends on $N_k$ and $h_k$ [22]. $\qquad\square$

# B  Algorithm Details for Optimizing Histogram Bins

We further discuss the detail process of finding the number of histogram bins that minimizes the MDL score, for which the pseudo-code is provided in Algorithm 3.

While the search space for the number of bins, denoted as $h$, for the histogram can range from 1 to the number of data points, this is computationally expensive in practice. To mitigate this, we first narrow down the range of $h$ by fixing a step size $g$ and searching among $h \in \{1, g + 1, 2g + 1, ...\}$. The MDL-score will initially decrease as we increase the number of bins, since the the goodness-of-fit of the histogram increases. However, when the number of bins become too large, the MDL-based regularization terms will start to dominate, and hence the MDL-score starts to increase.

Assuming the MDL-score starts to increase at $h = eg + 1$ (where $e$ is a positive integer), we can then narrow the range to $(e - 2)g + 1 \le h \le eg + 1$, since the maximum so far is reached at $(e - 1)g + 1$.

Finally, we iterate over all $h$ within this narrowed range, and we select the number of bins with the smallest MDL score.

# C  Full Experiment Details for Reproducibility

## C.1  Experiment compute resources

The runtimes reported in Section 6.5 for *all* algorithms are recorded on the CPU machines with the AMD EPYC 7702 cores.

**Algorithm 3** Learn the MDL-optimal histogram

---

**Input:** Target values $y^{\{S\}}$ covered by a leaf node $S$, step size $g$
**Output:** Histogram $H_S$ that minimizes the MDL-score

---

**28** $h' \leftarrow 1; best\_score \leftarrow +\infty$
**29** **while** *True* **do**
**30**     $mdl \leftarrow$ calculate the MDL-score given $h'$
**31**     **if** $mdl < best\_score$ **then**
**32**        $best\_score \leftarrow mdl$
**33**        $h' \leftarrow h' + g$
**34**     **else**
**35**        break
**36**     **end**
**37** **end**
**38** $bound\_low \leftarrow \max(1, h' - 2g)$ ;         `// Since `$h - 2g$` may be smaller than 1`
**39** $bound\_high \leftarrow h'$
**40** $h^* \leftarrow$ the optimal $h$ within the range $bound\_low \leq h \leq bound\_high$ that minimizes the MDL-score
**41** **return** $h^*$

---

### C.2  Data preparation

We noticed that duplicated values in datasets may cause some of our competitor methods to fail. For instance, for those methods that adopt the Gaussian kernel, duplicated values can lead to the standard deviation estimated as 0 for some subsets of data points). Thus, we add a very small noise generated by $N(0, 10^{-3})$ to all data columns, *which are used for all methods*.

### C.3  Parameters for CDTree

**The parameter $C$.**  We first discuss the parameter $C$ that controls the hierarchical structure for the search space of the splitting values, for which we set $C = 5$ in our experiments.

The parameter $C$ can be used to express a prior belief on the candidate splitting values. For instance, when $C = 5$, the code length needed for encoding the 5 quantiles when $d = 1$ are the same. Equivalently, the prior probabilities of these 5 quantiles are the same.

As these 5 quantiles divide the values into 6 subsets, essentially, we are saying that the first quantile among these 5 quantiles, i.e., the $1/6$-quantile, and the third quantile, i.e., the $1/2$-quantile ($3/6$-quantile), have the same prior probability.

However, if we set $C = 1$, there will be only 1 quantile when $d = 1$, i.e., the $1/2$-quantile. In this case, the prior probability for the $1/2$-quantile will be different (larger) than the $1/6$-quantile.

Thus, the parameter $C$ can be used to express the prior belief in the splitting values. That is, one can ask herself that, is it equally likely, *a priori*, to observe a node containing the splitting value equal to the $1/100$-quantile, and to observe a node containing the splitting value equal to the $1/2$-quantile (i.e., the median)? If the answer is yes, $C$ might be set as 99. Otherwise, a smaller $C$ should be considered.

**The step size $g$.**  We set $g = 30$ in searching the number of histogram bins in Algorithm 3. Note that if we set $g$ very large, we only need very few iterations to "narrow" down the range for the number of bins for histograms; nevertheless, the resulting range will not be very narrow in this case. On the other hand, if we set $g$ very small, it may cost a lot of time to obtain the narrower range. In that sense, $g = 30$ seems a rather balanced choice.

**The histogram boundary.**  Further, as discussed in Section 7, we assume that the range of the target variable for each dataset is known. Thus, we set the global range of the histograms based on the range of the full dataset (before train/test split in the cross-validation) plus/minus a small constant, chosen as $10^{-3}$.

### C.4 Details for competitor algorithms

We use the implementation of CKDE, NKDE, LSCDE, NF, and MDN from from the Python package "cde",[1] [46] and the implementation of LinCDE and CADET from the original authors.

**CADET.** No specific guidelines for regularization is provided in the original paper [6]; further, in the author's original implementation, the standard regularization based on the tree size is not available. Instead, we can only choose the regularization mode in 'constant', 'AIC', and 'BIC'. We pick the 'BIC' as it behaves similarly to MDL asymptotically [15].

**CKDE.** The bandwidth is tuned by optimizing the cross-validation maximum likelihoods, which is implemented in 'statsmodels' [48], the standard Python package for multivariate statistical analysis, and directly used in the 'cde' package [46]. For the three largest datasets ('support2', 'navalprop', and 'supercond'), we fail to obtain the cross-validation tuning results within 10 hours for a single fold, and hence we adopt the 'normal reference' for the bandwidth selection.

Note that it is not required to specify the range for searching the bandwidth, as the implementation is essentially based on the 'optimize' function (for continuous optimization) in the well-known Python package 'scipy' [56].

**NKDE.** The bandwidth and the $\epsilon$ (the parameter that controls the neighbor range) are tuned by optimizing the cross-validation maximum likelihoods, which is implemented in the Python package 'cde' [46]. Similar to CKDE, the search space for these two parameters are not required. For two very large datasets ('support2' and 'supercond'), we fail to obtain the cross-validation tuning results within 10 hours for a single fold, and hence we adopt the 'normal reference' for the bandwidth selection.

**CART-k and CART-h.** In these methods, a CART regression tree is first trained, with the regularization hyperparameter for pruning tuned by cross-validation. After fixing the tree structure, a kernel density estimation model and a histogram model are fit to the subsets of data points on each leaf node for CART-k and CART-h, respectively. The bandwidth for the former is tuned by cross-validation, while the number of bins for the latter is picked to optimize the MDL score (given the fixed tree structure). Specifically, we use the CART implementation from the scikit-learn [34] Python package.

**LSCDE.** No tuning for bandwidth is discussed in the original paper, nor implemented. Thus, we stick to the default setting.

**LinCDE.** The tree depth is set as 3, which is a common setting for training boosted tree models.

**NF and MDN.** As suggested by Rothfuss et al. [45], we use 'noise regularization' for both NF and MDN. Specifically, we choose the standard deviation of added noise for both the features and the target as $0.01$. We further noticed that adding the standard dropout with dropout rate equal to $0.1$ gives more stable results. Other hyperparameters (e.g., the number of hidden layers) are kept as default.

## D More experiment results

### D.1 Standard deviation of negative log-likelihoods

In Table 3, we report the negative log-likelihoods together with the standard deviations obtained by the five-fold cross-validation. In comparisons to other competitors, we observe no anomalously larger standard deviations for our method.

### D.2 Robustness against irrelevant features

In Figure 5, we show the results that demonstrate the stability of the negative log-likelihoods when independent gaussian features are added to the 14 UCI datasets, in which we observe similar results as in Figure 4, i.e., our proposed method CDTree is extremely stable across all datasets except for the very small dataset 'slump'.

Table 3: Negative log-likelihoods (smaller is better) on test sets, together with the standard deviation, obtained using five-fold cross-validation.

| | Interpretable models | | | | | | | Black-box models | | |
|---|---|---|---|---|---|---|---|---|---|---|
| dataset | CADET | CART-h | CART-k | CKDE | LSCDE | NKDE | Ours | LinCDE | MDN | NF |
| energy | 3.55 (0.22) | 3.09 (0.01) | 3.06 (0.17) | 2.47 (0.02) | 3.38 (0.01) | 3 (0.05) | 2.93 (0.01) | 2.93 (0.01) | 2.78 (0.02) | 2.86 (0.05) |
| synchronou | -2.93 (0.24) | -1.63 (0.07) | -1.86 (0.1) | -3.59 (0.09) | -1.25 (0.02) | -1.57 (0.04) | -2.11 (0.02) | -1.85 (0.03) | -2.94 (0.05) | -2.64 (0.15) |
| localizati | -0.23 (0.86) | -0.55 (0.29) | -0.01 (1.63) | -0.26 (1.79) | -0.61 (0.76) | -0.28 (0.98) | -0.66 (0.27) | -0.95 (0.11) | -0.68 (0.46) | -0.43 (0.95) |
| toxicity | 1.8 (0.19) | 1.5 (0.09) | 1.38 (0.09) | 1.32 (0.05) | 1.34 (0.02) | 1.55 (0.08) | 1.53 (0.09) | 1.29 (0.06) | 1.24 (0.08) | 1.23 (0.07) |
| concrete | 4.17 (0.47) | 3.75 (0.03) | 3.93 (0.57) | 3.32 (0.06) | 3.66 (0.03) | 3.91 (0.07) | 3.72 (0.06) | 3.47 (0.03) | 2.97 (0.06) | 3.18 (0.13) |
| slump | 3.42 (0.71) | 3.55 (0.19) | 3.43 (0.17) | 2.35 (0.2) | 2.91 (0.2) | 3.08 (0.25) | 3.34 (0.22) | 2.98 (0.15) | 2.23 (0.33) | 2.39 (0.18) |
| forestfire | 133.95 (106.29) | 3.96 (0.12) | 4.39 (1.35) | 4.85 (0.72) | 4.68 (0.14) | 5.55 (0.12) | 3.43 (0.19) | 4.35 (0.24) | 3.26 (0.48) | 3.23 (0.74) |
| navalpropo | -3.53 (0.12) | -3.3 (0.04) | -3.66 (0.07) | -2.8 (0) | -2.88 (0.01) | -3.19 (0.22) | -3.6 (0.06) | -3.36 (0) | -4.12 (0.04) | -3.75 (0.2) |
| skillcraft | 94.44 (46.08) | 0.46 (0.7) | -0.42 (0.53) | 1.54 (0.01) | 1.61 (0.02) | 1.56 (0.01) | -1.02 (0.03) | 1.26 (0.01) | 0.35 (1.26) | 1.11 (0.27) |
| sml2010 | 6.52 (2.65) | 2.85 (0.05) | 2.89 (0.43) | 1.61 (0.17) | 3.14 (0.01) | 3.12 (0.04) | 2.7 (0.06) | 2.97 (0.02) | 2.15 (0.02) | 2.61 (0.17) |
| thermograp | 2.21 (0.62) | 0.66 (0.05) | 0.72 (0.35) | 0.66 (0.06) | 0.94 (0.05) | 0.94 (0.05) | 0.64 (0.02) | 0.59 (0.06) | 0.56 (0.19) | 0.52 (0.03) |
| support2 | 97.33 (81.6) | 0.51 (0.04) | 0.32 (0.06) | 2.09 (0.01) | 2.46 (0.08) | 2.13 (0.02) | 0.29 (0.04) | 1.48 (0.07) | 1.53 (0.86) | 1.24 (0.22) |
| studentmat | 3.83 (0.39) | 2.65 (0.05) | 2.66 (0.05) | 2.89 (0.06) | 4.19 (0.41) | 3.11 (0.11) | 2.66 (0.07) | 2.59 (0.05) | 3.85 (0.34) | 3.54 (0.36) |
| supercondu | 9.6 (2.73) | 3.84 (0.01) | 4.36 (0.14) | 4.55 (0) | 4.17 (0.02) | 4.19 (0.15) | 3.48 (0.02) | 3.87 (0.02) | 3.33 (0.03) | 3.5 (0.04) |

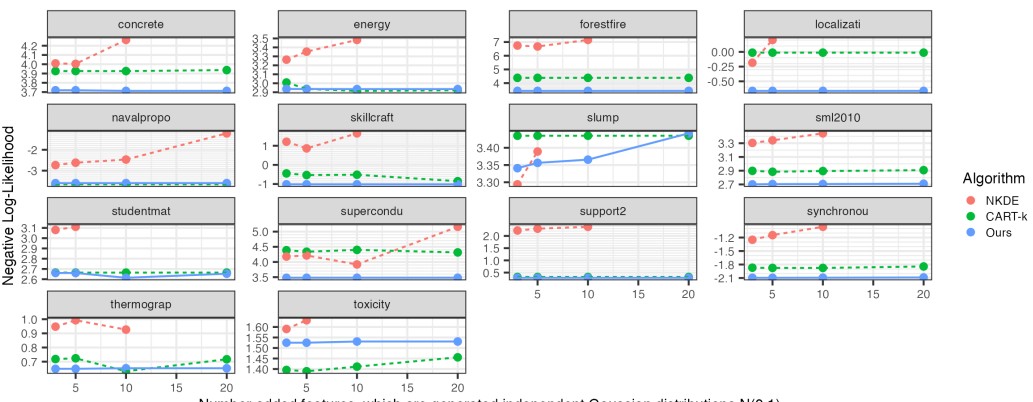

Figure 5: Negative-log-likelihoods with different number of added features, which are generated by independent Gaussian distributions. The results of CDTree, shown in blue solid lines, are extremely stable on all datasets expect for the very small 'slump' dataset.

# E   Medical costs visualizations

We show in Figure 6 the full results of applying CDTree to the medical expenses dataset used in Section 1. As a sanity check for the goodness-of-fit, we report the average cross-validation negative log-likelihood (on test sets) for CDTree is 9.03 (with the standard deviation 0.13), which is slightly better than that of the black-box boosted tree model LinCDE, with the average negative log-likelihood 9.55 (standard deviation 0.09). Each Histogram corresponds to a single leaf that describes a meaningful subgroup of patients, in which we observe 'age', 'BMI', and 'smoker' are the most distinguishing features to characterize each subgroup. Last, although the dataset contains 9 features and only 1338 samples, the illustrations show that CDTree does not require a very large dataset to reveal meaningful and comprehensible information.

---

[1] https://github.com/freelunchtheorem/Conditional_Density_Estimation

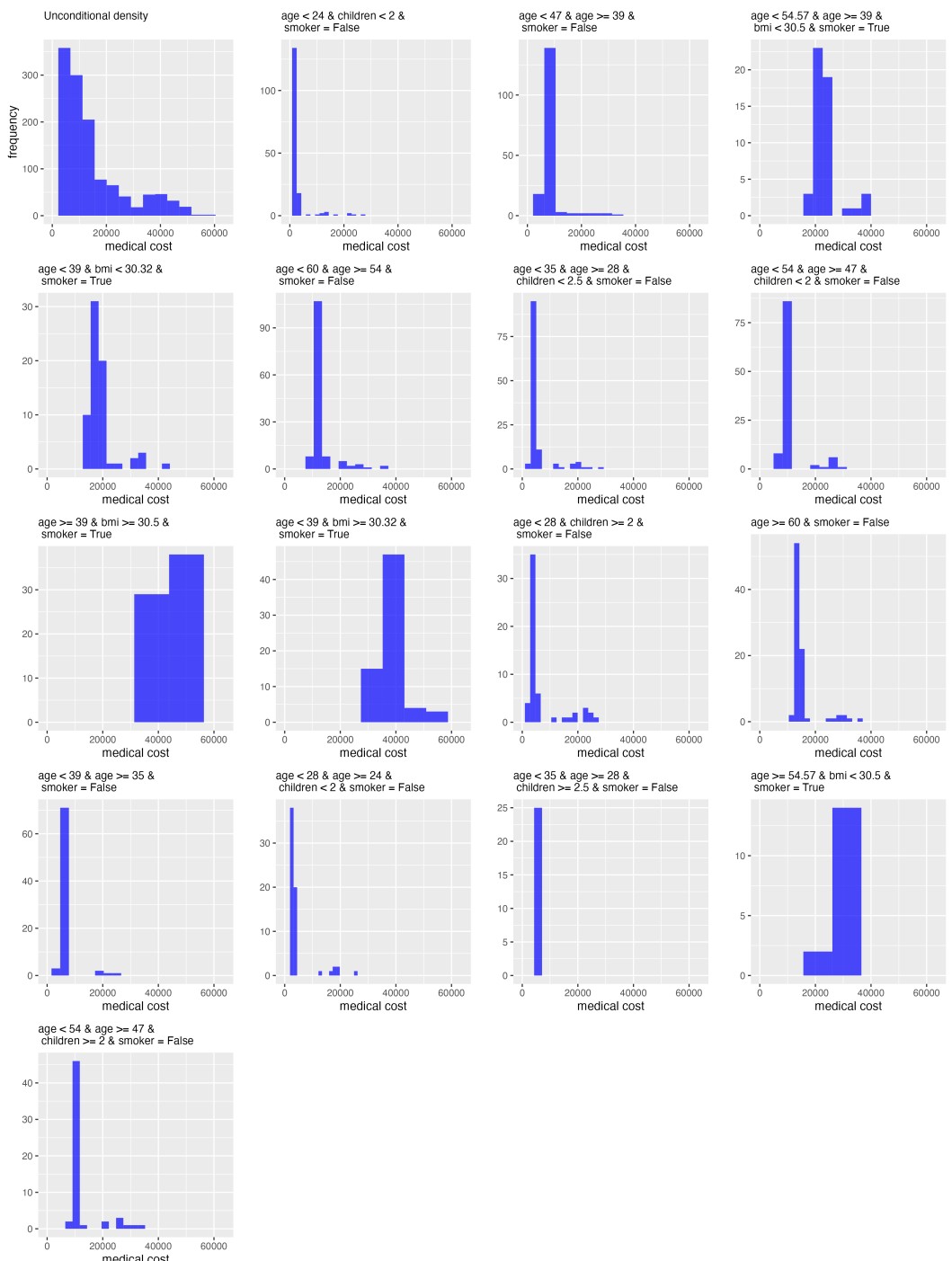

Figure 6: All leaves from the CDTree that describes the conditional density of medical costs, conditioned on demographic features. Root-to-leaf conditions are presented after removing logically redundant conditions.

