# OpenReview forum: "Conditional Density Estimation with Histogram Trees"
_NeurIPS.cc/2024/Conference — NeurIPS 2024 poster_

### Official Review · Reviewer_cqKh · 2024-06-23

**Soundness:** 3
**Presentation:** 3
**Contribution:** 3
**Rating:** 6
**Confidence:** 4

**Summary:**

This paper proposes a tree-based conditional density estimation model. Previously, conditional density estimation involved: 1) kernel density estimation-based method (requires expensive bandwidth tuning); 2) neural network (black-box approach); 3) a tree-based model which focuses on a Gaussian distribution for the target; 4) regression-based approach. In contrast to these approaches, the proposed method is 1) non-parametric; 2) uses a MDL principle which eliminates the need for hyperparameter tuning for regularization; 3) an iterative algorithm which searches for the optimal histogram in each leaf (thus modeling the full distribution). The authors claim that the proposed method is both more accurate (as measured by the log-loss) and more robust against irrelevant features. Further, the method is smaller in size and improves interpretability (which comes from a tree-based model). The experimental results are provided against a varying range of datasets from the UCI repository. The comparison is provided against other tree-based models and blackbox-approach methods.

**Strengths:**

1. Experimental results are comprehensive, providing superior accuracy and more compact model compared to other approaches.
2. The proposed method does not need extensive parameter tuning and is empirically fast.
3. The algorithm seems to be fairly novel and easy to understand.

**Weaknesses:**

1. The dataset used in the experiments is fairly small. To highlight empirical performance in training, the authors are advised to use larger datasets.
2. The method assumes that the support of the target variable is bounded.
3. Some presentational issues in the figure are present. See below in the Questions section.

**Questions:**

1. The method denoted as "Ours" should be formally labeled as "CDTree."
2. Are all of the attributes in the datasets continuous? How would the method handle discrete/categorical attributes?
3. What model selection criteria is used for CART models? Is it the same as Equation 4 which is used for CDTree?
4. Is it possible to move the pseudocodes into the main writeup?
5. Figure 3 may contain some presentational issues. What do "+", "circle" ,"square", etc mean?

**Limitations:**

Overall the paper is reasonable well-written. However, the experimental results are limited to fairly small datasets which unfortunately do not highlight potential scalability of the proposed method. While the focus of the method is on interpretability and accuracy, the authors would have to provide some empirical results on the training phase. In addition, please move the pseudocode description of the algorithm to the main writeup section.

---

> ### Author Rebuttal · Authors · 2024-08-06
>
> Dear reviewer,
>
> Thanks for the constructive advices! I will now reply to each point in the Weaknesses and Questions sections.
>
> # Regarding Weaknesses in the review:
> ## 1. Rebuttal to Weakness 1: In comparison to previous research for CDE, the datasets we used are NOT smaller.
> - We included 14 datasets in total, with various sample sizes (from 104 to 21263) and dimensionalities (from 5 to 82). We aim to test on both small and large datasets since 1) the sample size is often not too large in critical areas like healthcare; 2) (as mentioned in the Introduction) CDE is often used to understand the data collected from scienfitic experiments, in which data size is often small too, as data collection is expensive.
>
>    **Discussion about the data sizes in previous research:** for instance, in the most recent related work [12] *(Gao, Zijun, and Trevor Hastie. "LinCDE: conditional density estimation via Lindsey's method." Journal of machine learning research 23.52 (2022): 1-55.)*, which is a tree ensemble method, all datasets considered are below $2 000$, indicating that smaller datasets are indeed interesting to the CDE task. Further, the deep learning model proposed in [9] *(Dutordoir, Vincent, et al. "Gaussian process conditional density estimation." Advances in neural information processing systems 31 (2018)) {see page 8, figure 4}* considers benchmark datasets with similar sizes (number of rows) to ours (and the dimensionalities of datasets we consider are even in general higher).
>
> - Despite what has been discussed, we tried on a larger dataset about NYC taxi duraion (sample size $N=203686$), which we take from Kaggle. This is a dataset that can be used to predict the taxi trip durations based on the pick-up and drop-off locations (longitutde and latitude). We only use a subset of this data that corresponds to 1 month. However, most interpretable competitor methods fail on this dataset, and specifically **CKDE/NKDE/LinCDE fail to return any results because their (widely used) implementations caused that the memory limit is reached**.
>
>    Meanwhile, our method and CADET are the only interpretable methods that can be applied to this dataset. However, as shown in the table below, the negative log-likelihoods of our method are pretty close to those of the two neural network methods (NF and MDN), significantly outperforming the negative log-likelihoods of CADET. **Although the main goal of this paper is NOT to (just) introduce a CDE method that scales better than existing "shallow" models, this result indeed shows the potential of the scalability of CDTree to larger datasets**.
>
> | Methods | Mean (Negative Log-likelihoods)  | Standard Deviation |
> |-----------|-------|--------------------|
> | Ours (CDTree)     | 7.015 | 0.0038             |
> | CADET     | 8.290 | 0.0870             |
> | NF        | 6.921 | 0.1031             |
> | MDN       | 6.835 | 0.0056             |
>
>
> ## 2. Reply to Weakness 2:
> - We are aware of this limitation as we already discussed it in the Limitation of the paper.
>
> ## 3. Rebuttal to Weakness 3: Figure 3 does NOT contain the presentational issue mentioned in the review as the legend shows the meanings of different shapes.
> - As shown by the legend on the upper part of the figure, each method is shown by a different color TOGETHER with a different shape. We find that, in comparison to using different colors only (with the same shape), our visualization makes it easier to show the general picture for each method accross all datasets.
>
>
>
>
> # Reply to Questions:
> ## 1. The term "Ours" / "CDTree" for proposed method / model respectively
> We deliberately use “Ours” to refer to our method (the model we proposed, the learning criterion, and the algorithm, as a whole). By contrast, we use the term “CDTree'' to refer to the probabilistic model (learned from the data), which can be seen already from our abstract.
>
> ## 2. Our method can handle both discrete/continuous features.
> As discussed in Appendix B.2, we do not require the feature variables $X$ to be continuous. That is, while the target variable $Y$ needs to be continuous (otherwise there is no need for conditional density estimation), the feature variables can be both categorical and continuous. Just like the decision tree for classification/regression, the splitting conditions on the internal nodes can be defined for both discrete and continuous variables. We do assume that categorical variables are preprocessed and one-hot encoded though.
>
> ## 3. About CART model selection.
> The CART we use is the original version from the inventors. That is, the tree is first grown by iteratively maximizing the gini index, and then pruned by the so-called “cost complexity pruning”, i.e., by optimizing the impurity (MSE for regression) plus a regularization term defined as the number of nodes times a regularization hyperparameter, in which the hyperparameter is tuned by cross-validation in our experiment. These are (already) discussed in Appendix C.4.
> ## 4. Put (more) content about Algorithm in the main content.
> Yes, if the paper is accepted and then one more page is granted.
> ## 5. Resolved (see above).

---

> ### Comment · Reviewer_cqKh · 2024-08-12
> **Dear authors**
>
> (Mistake on the previous title - it was directed to the authors who provided the rebuttals):
> Thanks for your extensive reply on the points.
>
> Perhaps it is better to move clarifying points which are in the appendix to the main writeup (as I see that some answers to my questions are in the appendix which not many reviewers will have to look at, unless there is an explicit hint in the manuscript). I do not believe it will have much impact on the length of the manuscript. In addition, as you go through the draft once more, you will have opportunities to cut down redundant sentences/paragraphs and make your points more succinct.
>
> I still argue that Figure 3 is a very confusing figure - a visualization should not ask readers to combine a combination of color and shape. **Why are some of these shapes not aligned on the horizontal axis tick?** I also think there are too many horizontal ticks (yes, each tick = dataset) but there is not much space to put all of these information.
>
> **One more thing: I would also keep the fonts consistent in your reply - it actually does not help reviewers look at your reply in entirety and it looks less professional.**

---

> > ### Author Response · Authors · 2024-08-12
> > **Thanks for the further advices about the writing**
> >
> > Thanks for the further advices about the writing

---

> > ### Author Response · Authors · 2024-08-13
> > **Further clarifications for Figure 3**
> >
> > Dear reviewer,
> >
> > Thanks again for the detailed comments that help improve my paper further! Somehow the second half of your previous comments is missing in the notification email I received from "neurips2024-notifications", which did not contain your further comments about Figure 3. I only noticed this just now and hereby give further clarifications below.
> >
> > Although the common way of presenting the results in Figure 3 is to use a separate figure for each individual dataset (often with the x-axis representing "the number of added features" and the y-axis representing "the number of splits on irrelevant features"), this would lead to too many figures for our case (as we tested 14 datasets). We would need to put all these figures in the Appendix then. Thus, we chose to put the results of all datasets together, which can show that our method learns CDTrees with the number of irrelevant features very close to 0 across all datasets.  As a result, we just use the different shapes to keep the information about the number of features added for each "point" on the figure.
> >
> > I do agree that this figure may need more brainwork to process than other plots in the paper; however, the advantages are 1) avoiding too many figures for this single experiment subsection, and 2) using one plot for all datasets can better show the "general picture" of different methods. (Nevertheless, I would be happy to consider other possibilities in your mind as well. Thanks in advance!)
> >
> > Regarding the point you raised "Why are some of these shapes not aligned on the horizontal axis tick", we do this deliberately by "jittering" the points a bit, to avoid some points being fully covered by others.
> >
> > Last, thanks for advices about the fonts used in the replies as well!

---

### Official Review · Reviewer_n5cC · 2024-07-02

**Soundness:** 3
**Presentation:** 3
**Contribution:** 1
**Rating:** 5
**Confidence:** 5

**Summary:**

This paper addresses the problem of conditional density estimation, i.e. given a conditioning variable x estimate the whole distribution of y, with special emphasis on interpretability.
For this interpretability requirement, the authors resort to classical decision trees allowing to partition the conditioning space in an interpretable manner. The conditioned distributions of y are modelled with equally-spaced histograms with a variable number of bins.
This construction is optimised in an heuristic manner using an objective function that is built using a Minimum Description Length (MDL) approach, using different codes for the different parts (size and structure of the tree, splitting conditions and histograms) of the construction to be determined.
Experiments show log loss performance that is globally better than interpretable competitors on the selected datasets. Empirically, the trees obtained with this method are shallower than other interpretable tree-based methods and they are more robust to irrelevant features, in the sense that these are less often used for splitting.

**Strengths:**

The paper is generally  well written and easy to read.
The construction seems useful in practice as shown by the experiments and could be easily adopted by practitioners.

**Weaknesses:**

This work combines in a straightforward manner classical building blocks from the machine learning and information theory literature: decision trees, histograms, MDL, universal codes for integers, ... so there is little novelty in the construction

Although it is advertised that no hyperparameter tuning is needed thanks to MDL, there are still arbitrary choices for the priors and also the hyperparameter C (C=5 in the experiments).
For example, the tree size and structure could be encoded using Willems' approach in Context Tree Weighting  (preorder traversal of the tree, for each node use one bit to say if it is internal or leaf, which is equivalent to putting a 1/2 prior probability of splitting a node).
(F. M. J. Willems, Y. M. Shtarkov, and T. J. Tjalkens. The context-tree weighting method: Basic properties. Information Theory, IEEE Transactions on, 41(3):653–664, 1995.)
This should be discussed.

The MDL approach used in this work is called "crude" in [14]  since there is some arbitrariness in the choice of priors instead of some optimality criterion.
The only part in this work that is "refined" [14] is the NML  used for the histograms themselves but this is something that is well know (Proposition 1 is a simple extension of existing results).
This should be discussed.

The optimisation method is heuristic and thus its interest for the NeurIPS audience is limited (in fact, most of the algorithm is in the appendix, which indicates that it is not an important contribution).

For these reasons, I think that the novelty and the significance for NeurIPS is quite limited.

In my opinion, this work would fit better in a more applied venue.

Minor issues:
citations to books (e.g. [6,14]) should be more precise

line 186: "variable name" can be misleading, I suggest to say "variable index"

typo: line 231 "we iterative"

**Questions:**

paragraph explaining the role of C and d  (lines 191 to 194) is a bit confusing, can you provide an example showing what are the splitting options that are encoded when d>1 ?

**Limitations:**

Limitations were properly discussed.

---

> ### Author Rebuttal · Authors · 2024-08-06
>
> Thank you for your detailed comments and advice.
>
> We now address **each paragraph in the "Weakness" section** of your review.
>
> ## Factual Error in Paragraph 2:
> - The statement in the review that "no hyperparameter tuning is needed" is incorrect. Our paper highlights that the advantage of using MDL is the elimination of the need for a REGULARIZATION hyperparameter. We specifically use the term **"regularization hyperparameter"/"hyperparameter for regularization"** throughout the paper, in which we mentioned the word "hyperparameter" 4 times in total. We specifically emphasized this point because the “standard” way of regularization in decision tree learning is to penalize on the number of nodes times a "regularization hyperparameter", which needs extensive parameter tuning.
>
> ## Rebuttal about the statement that we are using "crude" MDL (Paragraph 3):
> - The claim that our MDL approach is "crude" is incorrect according to Chapter 14.3 of the cited book [14]. **See page 426. Here I quote: "In the end, we encode data D using a hybrid two-part/one-part universal model, explicitly encoding the models we want to select between and implicitly encoding any distributions contained in those models", which defines our approach as “the refined MDL”**. Additionally, while the NML regret for histograms is well-known, our contribution extends these results to a supervised setting with histograms and demonstrates that the NML regret for the CDTree is the product of the regret terms for all histograms on all leaves.
>
> ## Rebuttal about choosing priors (Paragraph 2):
> - Although there is indeed some flexibility in choosing the model encoding scheme within the MDL framework, similar to selecting priors in Bayesian model selection, encoding the model is not arbitrary but follows requirements and guidelines. This includes ensuring the prior probability defined on the model class sums to 1, preventing an excessively large penalty term that would require larger sample sizes to converge to the true model. The encoding scheme you mentioned is suboptimal because the corresponding prior probability is not proper (consider, e.g., the code word that corresponds to the case that every node is a leaf node, which cannot form a decision tree). Notably, many “old” methods that leverage MDL used various intuitive yet “crude” encoding schemes like this one, and it is hardly possible to review all of them (e.g., the one used in the C4.5 decision tree is different from the one you mentioned above).
>
> ## About choosing hyperparameter C (Paragraph 2):
> - The hyperparameter C establishes a hierarchical structure on the search space of continuous feature variables, setting levels for granularity. This hierarchy helps express prior beliefs about model complexity. For instance, intuitively, if a decision tree gives a split condition with a highly precise value like $X > 1.0001$, domain experts may ask, is this level of granularity (up to the 4 decimal places) necessary? Why not just $X > 1$? Thus, from the perspective of model selection, the former condition is "more complex" (more bits required to encode) than the latter one. However, as different feature variables may have very different ranges, instead of considering the granularity in terms of the decimal precision, in this paper we consider the granularity in terms of the quantiles (e.g., splitting on the 1/2-quantile may be considered "simpler" than splitting on the 1/10000-quantile). As explained in Appendix C.2, just like we have different numeral systems (e.g., decimal/binary), which set up different hierarchical structures for the levels of granularities, the hyperparameter C plays a similar role here in setting up the hierarchical structures for the quantiles.
> - We have shown that a global ad-hoc choice for $C = 5$ works well for **all datasets in our experiments section**. By contrast,  it is impossible to have a single value for the regularization hyperparameter used in traditional decision tree learning that works well for different datasets, which also does not carry this natural meaning of the $C$ in our paper.
>
> ## About our algorithimic innovations (Paragraph 4):
> - While our algorithm is greedy, it addresses the challenging task of optimizing the tree structure and separate models (on leaves) simultaneously. Recent algorithmic advancements for classification/regression trees have been reviewed, highlighting the challenges for CDE tasks, which motivates the choice for the heuristic algorithm.
> - Our main algorithmic innovations are described in Section 5 (not the appendix), including 1) no pruning for the tree, which favors the model complexity, and 2) searching the histograms directly (without using any "guessing" heuristics that are common in traditional "model tree" methods). We also include in Section 5 a high-level yet complete description of the algorithm process. The pseudo-code and other details are put to the appendix for reproducibility due to the page limit.
> - Due to the fact that improving on the heuristic algorithms for classification/regression trees often require a whole research paper to describe (as reviewed in our paper), we consider further improving on our proposed algorithm for CDE as future research.
>
> ## Rebuttal about the characterization of our method as a "straightforward combination of classical building blocks".
> We respectfully disagree on this statement as it overlooks our contributions:
> - We propose the first single tree-based model specifically designed for the CDE task (as CADET essentially used the same model as CART due to the equivalence between the MSE loss and the Gaussian assumption);
> - We are the first to formalize the decision tree learning under the (modern) MDL framework as a model selection problem, introducing new encoding schemes for data (with NML) and the decision tree itself.
> - Our extensive experiments demonstrate that with CDTree, kernel-based methods are no longer the only choice for "shallow" models for CDE.

---

> > ### Comment · Reviewer_n5cC · 2024-08-08
> > **About "Factual Error in Paragraph 2" + "choosing priors" +  "choosing hyperparameter C"**
> >
> > Thank you for your detailed response.
> >
> > Your choice of prior for the model consists in encoding the size (number of leaves $K$) with Rissanen's code for integers and then a uniform prior over all trees with K leaves ($1/C_K$).
> > I mentioned another possibility which is the prior used in CTW, which puts 1/2 probability on splitting each node.
> > In a series of papers on Bayesian Context Trees, a more general version is considered with a parameter $\beta$ defining the probability of not splitting, that is, the larger beta, the larger the penalization of more complex models.
> > See Section 3.1 of
> >
> > *Papageorgiou, I., & Kontoyiannis, I. (2024). Posterior representations for Bayesian Context Trees: Sampling, estimation and convergence. Bayesian analysis, 19(2), 501-529.*
> >
> > I don't understand your remark:
> > *The encoding scheme you mentioned is suboptimal because the corresponding prior probability is not proper (consider, e.g., the code word that corresponds to the case that every node is a leaf node, which cannot form a decision tree)*
> >
> > If every node is a leaf node => the tree has only one node (the root) => it's a valid decision tree and its prior probability is 1/2 with Willems' scheme.
> >
> > As mentioned in the reference above on Bayesian Context Trees, one can see the splitting process as a Galton-Watson process, proving that the prior is proper.
> >
> > As you say: "C establishes a hierarchical structure"  and "This hierarchy helps express prior beliefs about model complexity". Although I understand that empirically C=5 worked well for all the datasets considered, there is no theoretical proof showing that this is a universal constant that shouldn't be tuned.
> > So I still think that "no hyperparameter tuning is needed" should be at least toned down, since there could be datasets for which some tuning would be beneficial.
> >
> > Can you clarify how your prior satisfies (1.) of page 426 of the MDL book?

---

> ### Author Response · Authors · 2024-08-11
> **More rebuttal**
>
> Dear reviewer,
>
> Thanks for the quick and detailed response. We have further rebuttals as follows.
>
> ## Further clarification about the prior you proposed
> - **One factual error**: the paper you cited, *(Papageorgiou, I., & Kontoyiannis, I. (2024). Posterior representations for Bayesian Context Trees: Sampling, estimation and convergence. Bayesian analysis, 19(2), 501-529.)*, considers $m$-ary trees, while in our paper we consider (full) binary trees only, as describe in Lines 104-105 and then emphasized in Line 177.
> - **Further explanations about why the prior probabilities you proposed do not sum to 1**. For simplicity, consider the case when a decision tree has 3 nodes in total, for which apparently only one possible tree structure exists (i.e., one root node and two leaf nodes). However, if you put a 1/2 prior probability on whether to split each node, this "only" structure has a prior probability of $(1/2)^3 < 1$. Hence, in this case, the sum of all prior probability masses is $(1/2)^3 < 1$ as well.
>
> ## Further clarification about the hyperparameter tuning
> - Again, as already mentioned in our previous rebuttal, we NEVER claimed that “no hyperparameter tuning is needed”; hence, I am not sure how we "tone it down".
> - We indeed introduced $C$, yet we have shown that a global ad-hoc choice for $C$ can work reasonably well for all datasets in our experiments (and whether tuning $C$ can further increase the predictive performance of CDTree is not related to the main research question in this paper). By contrast, this is hardly possible for the regularization hyperparameter; i.e., it is hardly possible to pick a single value with an intuitive meaning for the $\alpha$ in the traditional decision tree learning optimization function, often in the form of "$impurity + \alpha |T|$", in which $|T|$ denotes the size of the tree, and $impurity$ can be MSE for regression.
>
> ## Regarding Condition (1) of page 426 of the MDL book.
> - Notably, the text box on Page 426 does NOT give a rigorous definition of the "Refined MDL model selection", for the following reasons. First, see page 427, the 2nd paragraph, and here I quote *"The general idea is summarized in the box on page 426, which provides **something like a definition** of MDL model selection, **but only in a quite restricted context**. If we go beyond that context, these prescriptions cannot be used literally, but extensions in the same spirit suggest themselves."*. Second, the Condition (1.) of page 426 is also NOT a rigorous definition for the requirements of the priors, as the concept "quasi-uniform" is NOT defined throughout the book (see page 425, here I quote *"While we have **not formalized** what types of codes still count as quasi-uniform and what not ..."*). Hence, Condition (1) of page 426 is more like a general guideline summarized from the examples used in the book.
> - The last two paragraphs of page 425 (starting from "we can choose the code ..."; and note that the last paragraph of page 425 ends on the top of page 427) are a more general guideline for choosing the priors for model selection than the condition (1) of page 426. Specifically, our proposed prior is close to the general description given by the last sentence of the first paragraph on page 427; here I quote, "In some cases we can apply conditional codes which do achieve a “conditional” minimax regret, ...". That is, in general, we encode the model by
>
>        number of nodes => tree structure => node splitting conditions => number of histogram bins;
>
>    each step is a conditional uniform/quasi-uniform code (as the integer code is listed as an example of quasi-uniform in the book), conditioned on the value encoded in the previous step.
>
> - Notably, we quoted the content of page 427 (in our previous response) as it suffices to give the rebuttal to your previous statement that a mix of NML (one-part) and a model prior makes our MDL encoding "crude"; as shown on page 427, a mix of these two is by definition the form of the "refined" MDL model selection.

---

> > ### Comment · Reviewer_n5cC · 2024-08-12
> > **alternative prior and "factual error"**
> >
> > In *Papageorgiou, I., & Kontoyiannis, I. (2024). Posterior representations for Bayesian Context Trees: Sampling, estimation and convergence. Bayesian analysis, 19(2), 501-529.*,  **proper  $m$-ary trees** are considered,  which are defined as follows:
> >
> > *A tree $T$ is called proper if any node in $T$ that is not a leaf has exactly $m$ children.*
> >
> > For $m=2$, it corresponds to full binary trees.
> > So, **I don't see where the "factual error" is.**
> >
> > Regarding your futher explanation, I think I see where your misunderstanding comes from : in the Bayesian Context Trees prior,  the structure is encoded **without conditioning on the total number of leaves.**
> > The total number of leaves is not explicitly encoded as in your approach.
> > So, your example gives the correct prior of that particular tree but **it is not the only one**: you need to sum over the whole set of full binary trees.

---

> > > ### Author Response · Authors · 2024-08-12
> > > **Further clarification**
> > >
> > > Again, thanks for the detailed and quick response! (Nice to have these technical discussions anyway.)
> > >
> > >
> > > First, I never claimed that our prior is the only choice.
> > >
> > > Second, I looked at papers you cited and looked at the exact formula of the prior you proposed. Specifically,
> > >
> > > [A] *Papageorgiou, I., & Kontoyiannis, I. (2024). Posterior representations for Bayesian Context Trees: Sampling, estimation and convergence. Bayesian analysis, 19(2), 501-529. *,
> > >
> > > and [A] cited the below
> > >
> > > [B] *Kontoyiannis, Ioannis, et al. "Bayesian context trees: Modelling and exact inference for discrete time series." Journal of the Royal Statistical Society Series B: Statistical Methodology 84.4 (2022): 1287-1323.*
> > >
> > > And in [B], Lemma 1 shows that the prior $\pi(T) = \alpha ^ {|T| - 1} \beta ^ {|T| - L_D(T)}$ sums to 1, **for all trees $T$ with the depth smaller than the given number $D$, which does not apply to our case as we do not have the constraint on the tree depth**. (You could say, the max depth $D$ can be specified as the sample size $n$, yet this is a bit arbitrary as it has influence on the priors for trees that reach (or do not reach) the depth $D$ (see the formula). It is at least much less natural for the decision tree for CDE than for the variable-memory Markov chains in [A].
> > >
> > > Last, I found where another misunderstanding is from. In your original review, you stated that
> > >
> > > *"For example, the tree size and structure could be encoded using Willems' approach in Context Tree Weighting (preorder traversal of the tree, for each node use one bit to say if it is **internal or leaf**, which is equivalent to putting a 1/2 prior probability of splitting a node)"*. However, this is **different** from what is described in Section 3.1 of [A], as no prior probability will be put on those nodes that reach the max depth **(i.e., leaves)**. In fact, it seems to me that your statement above actually says that, given a list of nodes (with a pre-defined order), we specify a prior probability 1/2 on whether to split **independently** to each node; however, in a branching process there is no such indepdency. This is where the misunderstanding mainly comes from.
> > >
> > > It is indeed an elegant way to encode a tree with the max depth given. Very nice to know.

---

> > > > ### Comment · Reviewer_n5cC · 2024-08-12
> > > >
> > > > Although in [A] the focus is on trees with max depth = D, the constraint can be easily removed. Consider the remark in [A] Sec 3.1 regarding the equivalence to a Galton-Watson process.
> > > > In the case of max depth = $D$, the branching process is stopped at generation $D$. In the unconstrained case $D=\infty$, it corresponds to a standard Galton-Watson process and therefore **still defines a probability distribution, i.e., adds up to 1.**
> > > > Notice also that in the second part of the original CTW work ( Willems, F. M. (1998). The context-tree weighting method: **Extensions**. IEEE Transactions on Information Theory, 44(2), 792-798. ), the case of  $D=\infty$ is considered.
> > > >
> > > > In conclusion, this way of encoding a tree is a valid alternative for your problem.

---

> > > > > ### Author Response · Authors · 2024-08-12
> > > > >
> > > > > Thanks for the further clarification! Could you clarify why and how a standard Galton-Watson process defines a probability distribution that adds to 1?

---

> > > > > > ### Comment · Reviewer_n5cC · 2024-08-13
> > > > > > **probability measure on the space of trees**
> > > > > >
> > > > > > You can find the answer in the references given in [A]: Athreya and Ney, 2004; Harris, 1963.
> > > > > > In particular, see Harris Chap VI, Sec 4 "The probability measure P".
> > > > > > As noted in (Athreya and Ney, 2004) page 3, Harris treats a more general case.
> > > > > >
> > > > > > You can find another statement in the first paragraph of the proposition in Sec. 3 of:
> > > > > > Neveu, J. (1986). Arbres et processus de Galton-Watson. In Annales de l'IHP Probabilités et statistiques (Vol. 22, No. 2, pp. 199-207).

---

> > > > > > > ### Author Response · Authors · 2024-08-13
> > > > > > >
> > > > > > > Many thanks. Page 3 of Athreya is clear already.
> > > > > > >
> > > > > > > I hence take back my rebuttal point that your proposed prior is not proper, and I agree that it could be an alternative to the current encoding. I have editted the paper to discuss this.
> > > > > > >
> > > > > > > Besides this point, I argue that the following points in the review have been addressed in my rebuttal (including my further clarifications in the discussion period):
> > > > > > >
> > > > > > > - The MDL encoding in this paper should not be called "crude", and the choice of priors is not arbitrary yet follows guidelines, e.g., according to the book [14].
> > > > > > >
> > > > > > > - We did not advertise "no hyperparameter tuning is needed" throughout the paper. Further, introducing $C$ has several advantages than the traditional "regularization hyperparameter" including: 1) an ad-hoc global choice $C=5$ has been shown to work reasonable well, and 2) $C$ carries an intuitive meaning.
> > > > > > >
> > > > > > > - While the judgement of novelty is subjective, I respectfully disagree on the characterization of our method as a "straightforward combination of classical building blocks", for reasons that have been discussed in my (initial) Rebuttal.
> > > > > > >
> > > > > > > Thanks again for the detailed reviews and inspiring discussions for the priors (especially the always swift responses during the discussion period).

---

> > > > > > > > ### Comment · Reviewer_n5cC · 2024-08-13
> > > > > > > >
> > > > > > > > Given the discussions and the points addressed in the rebuttal, I raise my score to 5.

---

### Official Review · Reviewer_yfPy · 2024-07-06

**Soundness:** 3
**Presentation:** 3
**Contribution:** 1
**Rating:** 5
**Confidence:** 3

**Summary:**

This paper proposed to use decision tree with leaves formed by histogram models for conditional density estimation to gain interpretability. Characteristics of the proposed method, along with the density estimation accuracy and run time, are evaluated with numerical experiments.

**Strengths:**

Extensive experiments are conducted to evaluate multiple aspects of the proposed method, including comparison of the density estimation accuracy with related methods, as well as computational efficiency and interpretability-related aspects.

**Weaknesses:**

The proposed method can be viewed as a conditional density estimator with partitions on both X and Y space. There is limited novelty as far as I read from this paper.

**Questions:**

- If p(y|x) changes smoothly with respect to x, is the proposed method capable of capturing such smoothness, given it is based on partition of X?
- How to prevent overfitting?

**Limitations:**

The authors discussed some limitations of their method. There is no potential negative societal impact of their work.
I suggest providing some discussion on the questions raised above.

---

> ### Author Rebuttal · Authors · 2024-08-06
>
> Dear Reviewer,
>
> Although I understand that the judgment about novelty can be a subjective matter, I would appreciate it if more information is provided regarding the motivation for this judgment, which would be helpful for me to improve the paper.
>
> Regarding your two questions:
> 1. I am not sure what exactly you mean by "capturing such smoothness". Could you give a formal definition or an example of "capturing smoothness"? In general, partition-based models (including histograms, decision trees, regression trees, etc) approximate smooth functions (e.g., decision boundaries) in a piecewise manner.
> 2. As elaborated in the paper, the overfitting is prevented by formalizing the learning problem as a MDL model selection problem.

---

> > ### Comment · Reviewer_yfPy · 2024-08-13
> >
> > Thanks for the rebuttal.
> > Regarding smoothness, I was asking about the way that the conditional density changes wrt x. There can be roughly two cases: (1) very heteroscedastic case, where p(y|x) can change drastically across x values, and (2) for x1\approx x2, p(y|x=x1) is close to p(y|x=x2), and the limiting scenario would be p(y|x) = p(y) regardless of x values.
> > CDE methods that partitions X is generally suitable for case 1, and I would like to hear how the proposed method deals with case 2.

---

> > > ### Author Response · Authors · 2024-08-13
> > >
> > > Dear reviewer,
> > >
> > > Thanks for the clarification for the questions. For case (2), i.e., for the case "$x1\approx x2$ then $p(y|x=x1)$ is close to $p(y|x=x2)$", $x_1$ and $x_2$ will be in the same leaf node, and both $p(y|x=x1)$ and $p(y|x=x2)$ will be estimated by a single density estimator (a histogram in our proposed method).
> > >
> > > In the extreme case that $p(y|x) = p(y)$ for all $x$, the decision tree will be learned to have only one node (the root node). This is in fact shown by our experiment in Section 6.4 "Robustness to irrelevant features", in which we show that features that are (conditionally) independent to $y$ won't be used for tree splitting.

---

> > > > ### Comment · Reviewer_yfPy · 2024-08-13
> > > >
> > > > Thanks for the response. I raise my score since my concern on the partition of X is partially addressed, although there does not seem to be sufficient evidence that the proposed method works under case (2).

---

> > > > > ### Author Response · Authors · 2024-08-14
> > > > >
> > > > > Dear reviewer,
> > > > >
> > > > > Thanks for the further comments and thanks a lot for raising the score.
> > > > >
> > > > > Regarding your point that "there does not seem to be sufficient evidence that the proposed method works under case (2)", I have brief further clarifications as follows.
> > > > >
> > > > > - As shown in our experiment 6.4, our method is very robust against irrelevant features. This indicates that, in the extreme case you mentioned, i.e., when all features are independent of the target variable $Y$, our method would rarely split on any feature; hence, our model would indeed estimate all $p(y|x)$ by a single $p(y)$.
> > > > >
> > > > > - After a CDTree is learned from data, and if two feature (vector) $x_1$ and $x_2$ fall into the same leaf node, our model by definition considers the probability distribution $p(Y|X = x_1)$ "close" to $p(Y|X = x_2)$, since we estimate both of them by the same histogram model (that is equipped to this single leaf node). Notably, determining whether $p(Y|X = x_1)$ is "close enough" to $p(Y|X = x_2)$ is then equivalent to determining whether $x_1$ and $x_2$ "should" belong to the same leaf node, which is also equivalent to determining, during the learning process, whether this leaf node should be further split to let $x_1$ and $x_2$ fall into two different leaf nodes instead. Thus, determining all these points is essentially related to the trade-off between model complexity and the estimation accuracy (for both $p(Y|X = x_1)$ and $p(Y|X = x_2)$), which is handled by our proposed MDL model selection approach. Specifically, our highly competitive experiment results on the accuracy of conditional density estimation (measured by the negative log-likelihoods) demonstrate that our proposed method handles well the trade-off.

---

### Official Review · Reviewer_QurN · 2024-07-15

**Soundness:** 4
**Presentation:** 3
**Contribution:** 3
**Rating:** 7
**Confidence:** 4

**Summary:**

This paper proposes a new conditional density estimation algorithm based on histogram trees.
The base model corresponds to a full binary tree, where each internal node is associated with a split of the feature space in one coordinate, and each leaf node is associated with a histogram density estimator for the response variable for the feature variable falling into the leaf node.
To find the best among all possible models that best fit the data, the proposed algorithm invokes the MDL principle, which avoids any hyperparameter tuning, to derive the model selection criterion.
Since the optimization is infeasible to be solved exactly over all possible models, the paper proposes a greedy algorithm.
Experiments support that the resulting estimator CDTree is competitive against the state-of-the-art models.

**Strengths:**

- The paper proposes a nice algorithm for an important problem based on basic principles. This seems to be a nice practical application of the MDL principle.
- Though the resulting estimator is a single tree-based conditional density estimator, which is inherently "interpretable" thanks to the tree structure, experimental results show that CDTree is competitive against the SOTA algorithm based on ensemble. This is quite surprising.
- Experimental results are sufficiently thorough to understand the pros and cons of the proposed method.

**Weaknesses:**

While I like the paper overall, the manuscript is not spotless, especially in terms of presentation, grammatical errors, and typographical mistakes.
I believe that after a careful revision of the paper, this would be a nice addition to the community. (Though the fit might be much better with an applied statistics journal.)

**Questions:**

**Suggestions**
- The definition of a conditional density tree with histogram in Section 3 seems to be incomplete. I was able to understand the complete model after I read Section 4.3. I believe that this should be properly defined as a model in Section 3, since without a proper definition Section 4.1 and 4.2 read awkward.
For example, what is the dimension of $x$ and do you assume $y\in\mathbb{R}$? Somehow this is not explicitly defined.
- And I think the sentence in lines 108-110 is quite confusing, in the sense that while the model $M$ may depend on the dataset $D$, a covariate $x$ needs not be from the dataset to evaluate the conditional density.
- In Section 4.1, the definition of MDL optimal model is rather abrupt as the "universal model" is defined only in Section 4.2. I believe that the NML code should be defined before (1) or at least there should be a pointer to (2) in that paragraph. The notation $P_M(.)$ is also misleading. Please consider $P_M(\cdot | \cdot)$.

**Questions**
- Why did the authors define $\hat{\theta}$ in Proposition 1 while it is not used?
- Why are CKDE and NKDE put under "interpretable models"? I was confused as the authors emphasized at multiple points that kernel-based methods are not interpretable.
- A silly baseline is to consider a Gaussian density estimator as CADET assumes in the proposed MDL criterion of CDTree. This would result in a faster algorithm as the search space is simple. I am curious how this simple baseline would work in practice, as this can help understand the benefits of using the flexible histograms in place of Gaussians and using MDL principle separately. I would appreciate additional experiments, if time permits.
- (Minor) Regarding the discussion on runtime: Can you provide a bit more quantitative remark, is possible? For example, the authors mention that CART is highly optimized while the proposed algorithm is not. In which part of CDTree could be more optimized?
- Please revise the bibliographic info. For example, "mdl" in the title of [41] should be capitalized.

**Limitations:**

The authors note its limitation regarding the boundedness assumption of target variable.
On top of that, the algorithm is greedy and there is no guarantee on closeness between the actual estimator and the optimal estimator defined by the learning criterion.

---

> ### Author Rebuttal · Authors · 2024-08-05
>
> Thank you so much for the detailed, constructive, and very helpful comments on my writing!
>
> ## Presentational issues fixed
>
> I have fixed the presentational issues and typos you mentioned. Among others, one extremely helpful comment is that "lines 108-110 are confusing": in these lines I wrote "... $f (y|x)$ as $f _ k (x)$ ... ", but it should actually be  "... $f (y|x)$ as $f _ k (y)$ ... ".
>
> ## Rebuttal about "model definition not complete"
>
> We respectfully disagree that "The definition of a conditional density tree with histograms in Section 3 seems to be incomplete", and you seem to indicate part of the model definition is in Section 4. However, Section 3 contains all information about how to calculate the probability/likelihood of data given a fixed conditional density tree (regardless on whether it is actually a good model), which is sufficient for defining a probabilistic model. In contrast, Section 4 is all about the definition of the model selection criterion.
>
> ## Rebuttal about "applied statistics journal may be a better fit"
>
> The machine learning community has been recently highly interested in both conditional density estimation (CDE) (e.g., the references [9, 37, 48, 12] in the paper are from NeurIPS/ICML/JMLR) and interpretability, yet interpretable CDE methods are neglected. Second, the lack of CDTree, as the counterpart to the regression tree and classification tree, hinders the development of XAI methods (e.g., local surrogate models) for CDE.
>
> ## Reply to your questions:
>
> - **Regarding the first and last point**: you are correct. Thanks so much again!
> - **Whether CKDE/NKDE is interpretable model**: CKDE/NKDE (and other kernel-based models) are currently the standard (if not the only) choices if someone is looking for a "shallow" model for CDE. We argued in the paper that these models are "arguably less interpretable to (single) trees", yet we are also aware that these models may be considered more interpretable than tree ensembles/neural networks. (After all, there is still no precise definition of "interpretable models". )
> - **About comparison to the baseline with Gaussian assumption**: The MDL criterion we proposed CANNOT be used for Gaussian assumptions, as it is known that the regret term for the NML will be infinity (see page 298 of the book [14], {Grünwald 2007, The minimum description length principle MIT press, Example 11.1, Chapter 11}). I agree it would be interesting to do some sort of this kind of comparison though, as kind of an ablation study for the histograms and a supplementary to CART-h in the paper. However, the closest related work I can think of is (Proença, Hugo M., et al. "Discovering outstanding subgroup lists for numeric targets using MDL." ECML PKDD 2020), but they 1) consider rule lists (not trees), 2) they consider subgroup discovery for numeric targets (so not regression or CDE) and 3) they use the Bayesian encoding, not NML (which is not possible as discussed).
> - **Runtime of CART**: it depends on the search space of the regularization hyper-parameter. In our experiments, we set the range by “ccp_alphas = np.logspace(-5, 2, 30)”, and for all datasets, a single run should not exceed 100 seconds from what I remember.

---

> > ### Comment · Reviewer_QurN · 2024-08-12
> >
> > I appreciate the authors' response.
> > Please consider revising the manuscript with the points raised in my review, as well as other reviews. Especially I found that the reviewer n5cC raised many interesting questions, which could help further improve the depth of the paper if properly answered.
> > I will keep my score as is, since I believe there is a sufficiently good contribution in this paper.

---

### Decision · Program_Chairs · 2024-09-25

**Decision:**

Accept (poster)

**Comment:**

The paper presents an interesting tree-based, interpretable, conditional density estimator. All the reviews seem to agree that the method is of practical utility and is an interesting application of MDL principle. Interpretability in CDE is indeed under-studied. In case the paper is accepted, I request the authors to incorporate all the changes suggested by the reviewers.